# Do Less, Get More: Streaming Submodular Maximization with Subsampling

**Moran Feldman**
Open University of Israel
moranfe@openu.ac.il

**Amin Karbasi**
Yale University
amin.karbasi@yale.edu

**Ehsan Kazemi**
Yale University
ehsan.kazemi@yale.edu

## Abstract

In this paper, we develop the first one-pass streaming algorithm for submodular maximization that does not evaluate the entire stream even once. By carefully sub-sampling each element of the data stream, our algorithm enjoys the tightest approximation guarantees in various settings while having the smallest memory footprint and requiring the lowest number of function evaluations. More specifically, for a monotone submodular function and a $p$-matchoid constraint, our randomized algorithm achieves a $4p$ approximation ratio (in expectation) with $O(k)$ memory and $O(km/p)$ queries per element ($k$ is the size of the largest feasible solution and $m$ is the number of matroids used to define the constraint). For the non-monotone case, our approximation ratio increases only slightly to $4p + 2 - o(1)$. To the best or our knowledge, our algorithm is the first that combines the benefits of streaming and subsampling in a novel way in order to truly scale submodular maximization to massive machine learning problems. To showcase its practicality, we empirically evaluated the performance of our algorithm on a video summarization application and observed that it outperforms the state-of-the-art algorithm by up to fifty-fold while maintaining practically the same utility. We also evaluated the scalability of our algorithm on a large dataset of Uber pick up locations.

## 1 Introduction

Submodularity characterizes a wide variety of discrete optimization problems that naturally occur in machine learning and artificial intelligence [2]. Of particular interest is submodular maximization, which captures many novel instances of data summarization such as active set selection in non-parametric learning [31], image summarization [40], corpus summarization [28], fMRI parcellation [37], ensuring privacy and fairness [21], two-stage optimization [34] and removing redundant elements from DNA sequencing [27], to name a few.

Often the collection of elements to be summarized is generated continuously, and it is important to maintain at real time a summary of the part of the collection generated so far. For example, a surveillance camera generates a continuous stream of frames, and it is desirable to be able to quickly get at every given time point a short summary of the frames taken so far. The naïve way to handle such a data summarization task is to store the entire set of generated elements, and then, upon request, use an appropriate offline submodular maximization algorithm to generate a summary out of the stored set. Unfortunately, this approach is usually not practical both because it requires the system to store the entire generated set of elements and because the generation of the summary from such a large amount of data can be very slow. These issues have motivated previous works to use streaming submodular maximization algorithms for data summarization tasks [1, 17, 32].

The first works (we are aware of) to consider a one-pass streaming algorithm for submodular maximization problems were the work of Badanidiyuru et al. [1], who described a $1/2$-approximation streaming algorithm for maximizing a monotone submodular function subject to a cardinality con-

Table 1: Streaming algorithms for submodular maximization subject to a $p$-matchoid constraint.

| Algorithm | Function | Approx. Ratio | Memory | Queries per Element | Reference |
|---|---|---|---|---|---|
| Deterministic | Monotone | $4p$ | $O(k)$ | $O(km)$ | [8] |
| Randomized | Non-monotone | $\frac{5p+2+1/p}{1-\varepsilon}$ | $O(\frac{k}{\varepsilon^2} \log \frac{k}{\varepsilon})$ | $O(\frac{k^2 m}{\varepsilon^2} \log \frac{k}{\varepsilon})$ | [8] |
| Deterministic | Non-monotone | $\frac{9p+O(\sqrt{p})}{1-\varepsilon}$ | $O(\frac{k}{\varepsilon} \log \frac{k}{\varepsilon})$ | $O(\frac{km}{\varepsilon} \log \frac{k}{\varepsilon})$ | [8] |
| Deterministic | Non-monotone | $4p + 4\sqrt{p} + 1$ | $O(k\sqrt{p})$ | $O(\sqrt{p}km)$ | [33][2] |
| Randomized | Monotone | $4p$ | $O(k)$ | $O(km/p)$ | This paper |
| Randomized | Non-monotone | $4p + 2 - o(1)$ | $O(k)$ | $O(km/p)$ | This paper |

straint, and the work of Chakrabarti and Kale [7] who gave a $4p$-approximation streaming algorithm for maximizing such functions subject to the intersection of $p$ matroid constraints. The last result was later extended by Chekuri et al. [8] to $p$-matchoids constraints. For non-monotone submodular objectives, the first streaming result was obtained by Buchbinder et al. [5], who described a randomized streaming algorithm achieving 11.197-approximation for the problem of maximizing a non-monotone submodular function subject to a single cardinality constraint. Then, Chekuri et al. [8] described an algorithm of the same kind achieving $(5p + 2 + 1/p)/(1 - \varepsilon)$-approximation for the problem of maximizing a non-monotone submodular function subject to a $p$-matchoid constraint, and a deterministic streaming algorithm achieving $(9p + O(\sqrt{p}))/(1 - \varepsilon)$-approximation for the same problem.[1] Finally, very recently, Mirzasoleiman et al. [33] came up with a different deterministic algorithm for the same problem achieving an approximation ratio of $4p + 4\sqrt{p} + 1$.

In the field of submodular optimization, it is customary to assume that the algorithm has access to the objective function and constraint through oracles. In particular, all the above algorithms assume access to a value oracle that given a set $S$ returns the value of the objective function for this set, and to an independence oracle that given a set $S$ and an input matroid answers whether $S$ is feasible or not in that matroid. Given access to these oracles, the algorithms of Chakrabarti and Kale [7] and Chekuri et al. [8] for monotone submodular objective functions are quite efficient, requiring only $O(k)$ memory ($k$ is the size of the largest feasible set) and using only $O(km)$ value and independence oracle queries for processing a single element of the stream ($m$ is the number of matroids used to define the $p$-matchoid constraint). However, the algorithms developed for non-monotone submodular objectives are much less efficient (see Table 1 for their exact parameters).

In this paper, we describe a new randomized streaming algorithm for maximizing a submodular function subject to a $p$-matchoid constraint. Our algorithm obtains an improved approximation ratio of $2p + 2\sqrt{p(p+1)} + 1 = 4p + 2 - o(1)$, while using only $O(k)$ memory and $O(km/p)$ value and independence oracle queries (in expectation) per element of the stream, which is even less than the number of oracle queries used by the state-of-the-art algorithm for monotone submodular objectives. Moreover, when the objective function is monotone, our algorithm (with slightly different parameter values) achieves an improved approximation ratio of $4p$ using the same memory and oracle query complexities, i.e., it matches the state-of-the-art algorithm for monotone objectives in terms of the approximation ratio, while improving over it in terms of the number of value and independence oracle queries used. Additionally, we would like to point out that our algorithm also works in the online model with preemption suggested by Buchbinder et al. [5] for submodular maximization problems. Thus, our result for non-monotone submodular objectives represents the first non-trivial result in this model for such objectives for any constraint other than a single matroid constraint. Furthermore, despite the generality of our algorithm for a $p$-matchoid constraint (which includes, in particular, a cardinality constraint, a single matroid constraint and an intersection of multiple matroids), the

approximation ratio that it achieves is the state-of-the-art for all the above special cases. For example, for a single matroid constraint, our algorithm achieves an approximation ratio of $3 + 2\sqrt{2} \approx 5.828$, which improves over the previous state-of-the-art 8-approximation algorithm by Chekuri et al. [8].

In addition to mathematically analyzing our algorithm, we also studied its practical performance and scalability in video summarization and location summarization tasks. We observed that, while our algorithm preserves the quality of the produced summaries, it outperforms the running time of the state-of-the-art algorithm by an order of magnitude. We also studied the effect of imposing different $p$-matchoid constraints on these applications. **Most of the proofs for the theoretical results are deferred to the Supplementary Material.**

## 1.1 Additional Related Work

The work on (offline) maximizing a monotone submodular function subject to a matroid constraint goes back to the classical result of Fisher et al. [16], who showed that the natural greedy algorithm gives an approximation ratio of 2 for this problem. Later, an algorithm with an improved approximation ratio of $e/(e-1)$ was found for this problem [6], which is the best that can be done in polynomial time [35]. In contrast, the corresponding optimization problem for non-monotone submodular objectives is much less well understood. After a long series of works [11, 13, 25, 36, 43], the current best approximation ratio for this problem is 2.598 [3], which is still far from the state-of-the-art inapproximability result of 2.093 for this problem due to [36].

Several works have considered (offline) maximization of both monotone and non-monotone submodular functions subject to constraint families generalizing matroid constraints, including intersection of $p$-matroid constraints [26], $p$-exchange system constraints [14, 45], $p$-extendible system constraints [15] and $p$-systems constraints [15, 16, 19, 30]. We note that the first of these families is a subset of the $p$-matchoid constraints studied by the current work, while the last two families generalize $p$-matchoid constraints. Moreover, the state-of-the-art approximation ratios for all these families of constraints are $p \pm O(\sqrt{p})$ both for monotone and non-monotone submodular objectives.

The study of submodular maximization in the streaming setting has been mostly surveyed above. However, we would like to note that besides the above-mentioned results, there are also a few works on submodular maximization in the sliding window variant of the streaming setting [9, 12, 44].

## 1.2 Our Technique

Technically, our algorithm is equivalent to dismissing every element of the stream with an appropriate probability and then feeding the elements that have not been dismissed into the deterministic algorithm of [8] for maximizing a monotone submodular function subject to a $p$-matchoid constraint. The random dismissal of elements gives the algorithm two advantages. First, it makes it faster because there is no need to process the dismissed elements. Second, it is well known that such a dismissal often transforms an algorithm for monotone submodular objectives into an algorithm with some approximation guarantee also for non-monotone objectives. However, besides the above important advantages, dismissing elements at random also have an obvious drawback, namely, the dismissed elements are likely to include a significant fraction of the value of the optimal solution. The crux of the analysis of our algorithm is its ability to show that the above-mentioned loss of value due to the random dismissal of elements does not affect the approximation ratio. To do so, we prove a stronger version of a structural lemma regarding graphs and matroids (see Proposition 10) that was implicitly proved by [42] and later stated explicitly by [8]. This proposition provides a mapping from the elements of the optimal solution to elements of the solution $S$ chosen by our algorithm. This mapping helps us to show that the value of the elements of the optimal solution that do not belong to set $S$ is not too large compared to the value of $S$ itself. In this way, the stronger lemma we prove translates into an improvement in the bound on the performance of the algorithm, which is not sufficient to improve the guaranteed approximation ratio, but fortunately, is good enough to counterbalance the loss due to the random dismissal of elements.

We would like to note that the general technique of dismissing elements at random and then running an algorithm for monotone submodular objectives on the remaining elements, was previously used by [15] in the context of offline algorithms. However, the method we use in this work to counterbalance the loss of value due to the random dismissal of *streaming* elements is completely unrelated to the way this was achieved in [15].

## 2 Preliminaries

In this section, we introduce some notation and definitions that we later use to formally state our results. A set function $f\colon 2^{\mathcal{N}} \to \mathbb{R}$ on a ground set $\mathcal{N}$ is *non-negative* if $f(S) \geq 0$ for every $S \subseteq \mathcal{N}$, *monotone* if $f(S) \leq f(T)$ for every $S \subseteq T \subseteq \mathcal{N}$ and *submodular* if $f(S) + f(T) \geq f(S \cup T) + f(S \cap T)$ for every $S, T \subseteq \mathcal{N}$. Intuitively, a submodular function is a function that obeys the property of diminishing returns, i.e., the marginal contribution of adding an element to a set diminishes as the set becomes larger and larger. Unfortunately, it is somewhat difficult to relate this intuition to the above (quite cryptic) definition of submodularity, and therefore, a more friendly equivalent definition of submodularity is often used. However, to present this equivalent definition in a simple form, we need some notation. Given a set $S$ and an element $u$, we denote by $S + u$ and $S - u$ the union $S \cup \{u\}$ and the expression $S \setminus \{u\}$, respectively. Additionally, the marginal contribution of $u$ to the set $S$ under the set function $f$ is written as $f(u \mid S) \triangleq f(S + u) - f(S)$. Using this notation, we can now state the above mentioned equivalent definition of submodularity, which is that a set function $f$ is submodular if and only if

$$f(u \mid S) \geq f(u \mid T) \quad \forall\, S \subseteq T \subseteq \mathcal{N} \text{ and } u \in \mathcal{N} \setminus T \ .$$

Occasionally, we also refer to the marginal contribution of a set $T$ to a set $S$ (under a set function $f$), which we write as $f(T \mid S) \triangleq f(S \cup T) - f(S)$.

A *set system* is a pair $(\mathcal{N}, \mathcal{I})$, where $\mathcal{N}$ is the ground set of the set system and $\mathcal{I} \subseteq 2^{\mathcal{N}}$ is the set of *independent* sets of the set system. A *matroid* is a set system which obeys three properties: (i) the empty set is independent, (ii) if $S \subseteq T \subseteq \mathcal{N}$ and $T$ is independent, then so is $S$, and finally, (iii) if $S$ and $T$ are two independent sets obeying $|S| < |T|$, then there exists an element $u \in T \setminus S$ such that $S + u$ is independent. In the following lines we define two matroid related terms that we use often in our proofs, however, readers who are not familiar with matroid theory should consider reading a more extensive presentation of matroids, such as the one given by [38, Volume B]. A *cycle* of a matroid is an inclusion-wise minimal dependent set, and an element $u$ is *spanned* by a set $S$ if the maximum size independent subsets of $S$ and $S + u$ are of the same size. Note that it follows from these definitions that every element $u$ of a cycle $C$ is spanned by $C - u$.

A set system $(\mathcal{N}, \mathcal{I})$ is a *$p$-matchoid*, for some positive integer $p$, if there exist $m$ matroids $(\mathcal{N}_1, \mathcal{I}_1), (\mathcal{N}_2, \mathcal{I}_2), \dots, (\mathcal{N}_m, \mathcal{I}_m)$ such that every element of $\mathcal{N}$ appears in the ground set of at most $p$ out of these matroids and $\mathcal{I} = \{S \subseteq 2^{\mathcal{N}} \mid \forall_{1 \leq i \leq m} S \cap \mathcal{N}_i \in \mathcal{I}_i\}$. A simple example for a 2-matchoid is $b$-matching. Recall that a set $E$ of edges of a graph is a $b$-matching if and only if every vertex $v$ of the graph is hit by at most $b(v)$ edges of $E$, where $b$ is a function assigning integer values to the vertices. The corresponding 2-matchoid $\mathcal{M}$ has the set of edges of the graph as its ground set and a matroid for every vertex of the graph, where the matroid $\mathcal{M}_v$ of a vertex $v$ of the graph has in its ground set only the edges hitting $v$ and a set $E$ of edges is independent in $\mathcal{M}_v$ if and only if $|E| \leq b(v)$. Since every edge hits only two vertices, it appears in the ground sets of only two vertex matroids, and thus, $\mathcal{M}$ is indeed a 2-matchoid. Moreover, one can verify that a set of edges is independent in $\mathcal{M}$ if and only if it is a valid $b$-matching.

The problem of maximizing a set function $f\colon 2^{\mathcal{N}} \to \mathbb{R}$ subject to a $p$-matchoid constraint $\mathcal{M} = (\mathcal{N}, \mathcal{I})$ asks us to find an independent set $S \in \mathcal{I}$ maximizing $f(S)$. In the streaming setting, we assume that the elements of $\mathcal{N}$ arrive sequentially in some adversarially chosen order, and the algorithm learns about each element only when it arrives. The objective of an algorithm in this setting is to maintain a set $S \in \mathcal{I}$ which approximately maximizes $f$, and to do so with as little memory as possible. In particular, we are interested in algorithms whose memory requirement does not depend on the size of the ground set $\mathcal{N}$, which means that they cannot keep in their memory all the elements that have arrived so far. Our two main results for this setting are given by the following theorems. Recall that $k$ is the size of the largest independent set and $m$ is the number of matroids used to define the $p$-matchoid constraint.

**Theorem 1.** *There is a streaming $(2p + 2\sqrt{p(p+1)} + 1)$-approximation algorithm for the problem of maximizing a non-negative submodular function $f$ subject to a $p$-matchoid constraint whose space complexity is $O(k)$. Moreover, in expectation, this algorithm uses $O(km/p)$ value and independence oracle queries when processing each arriving element.*

**Theorem 2.** *There is a streaming $4p$-approximation algorithm for the problem of maximizing a non-negative monotone submodular function $f$ subject to a $p$-matchoid constraint whose space*

*complexity is $O(k)$. Moreover, in expectation, this algorithm uses $O(km/p)$ value and independence oracle queries when processing each arriving element.*

## 3 Algorithm

In this section we prove Theorems 1 and 2. Throughout this section we assume that $f$ is a non-negative submodular function over the ground set $\mathcal{N}$, and $\mathcal{M} = (\mathcal{N}, \mathcal{I})$ is a $p$-matchoid over the same ground set which is defined by the matroids $(\mathcal{N}_1, \mathcal{I}_1), (\mathcal{N}_2, \mathcal{I}_2), \ldots, (\mathcal{N}_m, \mathcal{I}_m)$. Additionally, we denote by $u_1, u_2, \ldots, u_n$ the elements of $\mathcal{N}$ in the order in which they arrive. Finally, for an element $u_i \in N$ and sets $S, T \subseteq \mathcal{N}$, we use the shorthands $f(u_i : S) = f(u_i \mid S \cap \{u_1, u_2, \ldots, u_{i-1}\})$ and $f(T : S) = \sum_{u \in T} f(u : S)$. Intuitively, $f(u : S)$ is the marginal contribution of $u$ to the part of $S$ that arrived before $u$ itself.

Let us now present the algorithm we use to prove our results. This algorithm uses a procedure named EXCHANGE-CANDIDATE which appeared also in previous works, sometimes under the exact same name. EXCHANGE-CANDIDATE gets an independent set $S$ and an element $u$, and its role is to output a set $U \subseteq S$ such that $S \setminus U + u$ is independent. The pseudocode of EXCHANGE-CANDIDATE is given as Algorithm 1.

---

**Algorithm 1:** EXCHANGE-CANDIDATE $(S, u)$

1   Let $U \leftarrow \varnothing$.
2   **for** $\ell = 1$ **to** $m$ **do**
3     **if** $(S + u) \cap \mathcal{N}_\ell \notin \mathcal{I}_\ell$ **then**
4       Let $X_\ell \leftarrow \{x \in S \mid ((S - x + u) \cap \mathcal{N}_\ell) \in \mathcal{I}_\ell\}$.
5       Let $x_\ell \leftarrow \arg \min_{x \in X_\ell} f(x : S)$.
6       Update $U \leftarrow U + x_\ell$.

7   **return** $U$.

---

**Algorithm 2:** SAMPLE-STREAMING

1   Let $S_0 \leftarrow \varnothing$.
2   **for** *every arriving element* $u_i$ **do**
3     Let $S_i \leftarrow S_{i-1}$.
4     **with** *probability* $q$ **do**
5       Let $U_i \leftarrow$ EXCHANGE-CANDIDATE$(S_{i-1}, u_i)$.
6       **if** $f(u_i \mid S_{i-1}) \geq (1 + c) \cdot f(U_i : S_{i-1})$ **then** Let $S_i \leftarrow S_{i-1} \setminus U_i + u_i$.

7   **return** $S_n$.

---

Our algorithm, which uses the procedure EXCHANGE-CANDIDATE, is given as Algorithm 2. This algorithm has two parameters, a probability $q$ and a value $c > 0$. Whenever the algorithm gets a new element $u$, it dismisses it with probability $1 - q$. Otherwise, it finds using EXCHANGE-CANDIDATE a set $U$ of elements whose removal from the current solution maintained by the algorithm allows the addition of $u$ to this solution. If the marginal contribution of adding $u$ to the solution is large enough compared to the value of the elements of $U$, then $u$ is added to the solution and the elements of $U$ are removed. While reading the pseudocode of the algorithm, keep in mind that $S_i$ represents the solution of the algorithm after $i$ elements have been processed.

**Observation 3.** *Algorithm 2 can be implemented using $O(k)$ memory and, in expectation, $O(qkm)$ value and independence oracle queries per arriving element.*

The following technical theorem is the main tool that we use to analyze the approximation ratio of Algorithm 2; its proof can be found in Appendix A. Let $OPT$ be an independent set of $\mathcal{M}$ maximizing $f$, and let $A$ be the set of elements that ever appeared in the solution maintained by Algorithm 2—formally, $A = \bigcup_{i=1}^{n} S_i$.

**Theorem 4.** *Assuming $q^{-1} = (1 + c)p + 1$, $\mathbb{E}[f(S_n)] \geq \frac{c}{(1+c)^2 p} \cdot \mathbb{E}[f(A \cup OPT)]$.*

Proving our result for monotone functions (Theorem 2) is now straightforward.

*Proof of Theorem 2.* By plugging $c = 1$ and $q^{-1} = 2p + 1$ into Algorithm 2, we get an algorithm which uses $O(k)$ memory and $O(km/p)$ oracle queries by Observation 3. Additionally, by Theorem 4, this algorithm obeys

$$\mathbb{E}[f(S_n)] \geq \frac{c}{(1+c)^2 p} \cdot \mathbb{E}[f(A \cup OPT)] = \frac{1}{4p} \cdot \mathbb{E}[f(A \cup OPT)] \geq \frac{1}{4p} \cdot f(OPT) \ ,$$

where the second inequality follows from the monotonicity of $f$. Thus, the approximation ratio of the algorithm we got is at most $4p$. $\qquad\square$

Proving our result for non-monotone functions is a bit more involved and requires the following known lemma.

**Lemma 5** (Lemma 2.2 of [4]). *Let $g\colon 2^{\mathcal{N}} \to \mathbb{R}_{\geq 0}$ be a non-negative submodular function, and let $B$ be a random subset of $\mathcal{N}$ containing every element of $\mathcal{N}$ with probability at most $q$ (not necessarily independently), then $\mathbb{E}[g(B)] \geq (1-q) \cdot g(\varnothing)$.*

*Proof of Theorem 1.* By plugging $c = \sqrt{1 + 1/p}$ and $q^{-1} = p + \sqrt{p(p+1)} + 1$ into Algorithm 2, we get an algorithm which uses $O(k)$ memory and $O(km/p)$ oracle queries by Observation 3. Additionally, by Theorem 4, this algorithm obeys

$$\mathbb{E}[f(S_n)] \geq \frac{c}{(1+c)^2 p} \cdot \mathbb{E}[f(A \cup OPT)] \ .$$

Let us now define $g\colon 2^{\mathcal{N}} \to \mathbb{R}_{\geq 0}$ to be the function $g(S) = f(S \cup OPT)$. Note that $g$ is non-negative and submodular. Thus, by Lemma 5 and the fact that $A$ contains every element with probability at most $q$ (because Algorithm 2 accepts an element into its solution with at most this probability), we get

$$\mathbb{E}[f(A \cup OPT)] = \mathbb{E}[g(A)] \geq (1-q) \cdot g(\varnothing) = \frac{p + \sqrt{p(p+1)}}{p + \sqrt{p(p+1)} + 1} \cdot f(OPT)$$

$$= \frac{p + \sqrt{p(p+1)}}{\sqrt{1 + 1/p} \cdot (p + \sqrt{p(p+1)})} \cdot f(OPT) = \frac{1}{c} \cdot f(OPT) \ .$$

Combining the two above inequalities, we get

$$\mathbb{E}[f(S_n)] \geq \frac{f(OPT)}{(1+c)^2 p} = \frac{f(OPT)}{(2 + 2\sqrt{1 + 1/p} + 1/p)p} = \frac{f(OPT)}{2p + 2\sqrt{p(p+1)} + 1}$$

Thus, the approximation ratio of the algorithm we got is at most $2p + 2\sqrt{p(p+1)} + 1$. $\square$

## 4 Experiment

In this section, we investigate the performance of our algorithm on two data summarization applications. In the first part, we replicate the exact setting of Mirzasoleiman et al. [33] and compare the performance our algorithm in this setting with the performance of the algorithm of Mirzasoleiman et al. [33]. Unfortunately, to allow such a comparison we had to resort to the relatively small datasets that existing algorithms can handle. Interestingly, however, despite the small size of these datasets, we could still observe the superiority of our method against the state-of-the-art. In the second part, we investigate the scalability of our algorithm to larger datasets.

### 4.1 Video Summarization

In this section, we evaluate the performance of our algorithm (SAMPLE-STREAMING) on a video summarization task and compare it with SEQDPP [18][3] and LOCAL-SEARCH [33].[4] For our experiments, we use the Open Video Project (OVP) and the YouTube datasets, which have 50 and 39 videos, respectively [10].

Determinantal point process (DPP) is a powerful method to capture diversity in datasets [24, 29]. Let $\mathcal{N} = \{1, 2, \cdots, n\}$ be a ground set of $n$ items. A DPP defines a probability distribution over all subsets of $\mathcal{N}$, and a random variable $Y$ distributed according to this distribution obeys $\Pr[Y = S] = \frac{\det(L_S)}{\det(I+L)}$ for every set $S \subseteq \mathcal{N}$, where $L$ is a positive semidefinite kernel matrix, $L_S$ is the principal sub-matrix of $L$ indexed by $S$ and $I$ is the $n \times n$ identity matrix. The most diverse subset of $\mathcal{N}$ is the one with the maximum probability in this distribution. Unfortunately, finding this set is NP-hard [22], but the function $f(S) = \log \det(L_S)$ is a non-monotone submodular function [24].

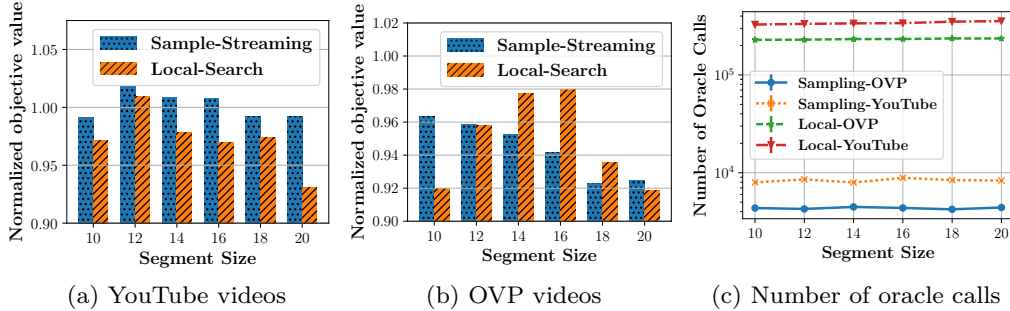

Figure 1: Comparing the normalized objective value and running time of SAMPLE-STREAMING and LOCAL-SEARCH for different segment sizes.

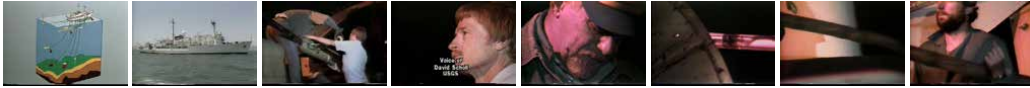

Figure 2: Summary generated by SAMPLE-STREAMING for OVP video number 60.

We follow the experimental setup of [18] for extracting frames from videos, finding a linear kernel matrix $L$ and evaluating the quality of produced summaries based on their F-score. Gong et al. [18] define a sequential DPP, where each video sequence is partitioned into disjoint segments of equal sizes. For selecting a subset $S_t$ from each segment $t$ (i.e., set $\mathcal{P}_t$), a DPP is defined on the union of the frames in this segment and the selected frames $S_{t-1}$ from the previous segment. Therefore, the conditional distribution of $S_t$ is given by, $\Pr[S_t|S_{t-1}] = \frac{\det(L_{S_t \cup S_{t-1}})}{\det(I_t + L)}$, where $L$ is the kernel matrix defined over $\mathcal{P}_t \cup S_{t-1}$, and $I_t$ is a diagonal matrix of the same size as $\mathcal{P}_t \cup S_{t-1}$ in which the elements corresponding to $S_{t-1}$ are zeros and the elements corresponding to $\mathcal{P}_t$ are 1. For the detailed explanation, please refer to [18]. In our experiments, we focus on maximizing the non-monotone submodular function $f(S_t) = \log \det(L_{S_t \cup S_{t-1}})$. We would like to point out that this function can take negative values, which is slightly different from the non-negativity condition we need for our theoretical guarantees.

We first compare the objective values (F-scores) of SAMPLE-STREAMING and LOCAL-SEARCH for different segment sizes over YouTube and OVP datasets. In each experiment, the values are normalized to the F-score of summaries generated by SEQDPP. While SEQDPP has the best performance in terms of maximizing the objective value, in Figures 1(a) and 1(b), we observe that both SAMPLE-STREAMING and LOCAL-SEARCH produce summaries with very high qualities. Figure 2 shows the summary produced by our algorithm for OVP video number 60. Mirzasoleiman et al. [33] showed that their algorithm (LOCAL-SEARCH) runs three orders of magnitude faster than SEQDPP [18]. In our experiments (see Figure 1(c)), we observed that SAMPLE-STREAMING is 40 and 50 times faster than LOCAL-SEARCH for the YouTube and OVP datasets, respectively. Note that for different segment sizes the number of frames remains constant; therefore, the time complexities for both SAMPLE-STREAMING and LOCAL-SEARCH do not change.

In a second experiment, we study the effect of imposing different constraints on video summarization task for YouTube video number 106, which is a part of the America's Got Talent series. In the first set of constraints, we consider 6 (for 6 different faces in the frames) partition matroids to limit the number of frames containing each face $i$, i.e., a 6-matchoid constraint[5] $\mathcal{I} = \{S \subseteq \mathcal{N} : |S \cap \mathcal{N}_i| \le k_i\}$, where $\mathcal{N}_i \subseteq \mathcal{N}$ is the set of frames containing face $i$ for $1 \le i \le 6$. For all the $i$ values, we set $k_i = 3$. In this experiment, we use the same methods as described by Mirzasoleiman et al. [33] for face recognition. Figure 3(a) shows the summary produced for this task. The second set of constraints is a 3-matchoid, where matroids limit the number of frames containing each one of the three judges. The summary for this constraint is shown in Figure 3(b). Finally, Figure 3(c) shows a summary with a single partition matroid constraint on the singer.

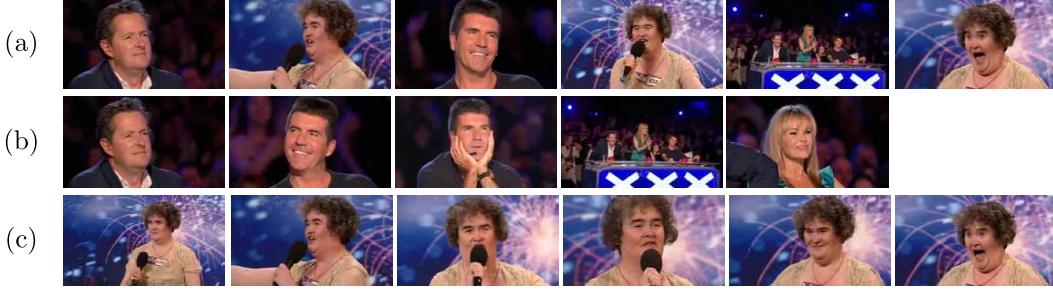

Figure 3: Summaries generated by SAMPLE-STREAMING for YouTube video number 106: (a) a 6-matchoid constraint, (b) a 3-matchoid constraint and (c) a partition matroid constraint.

## 4.2 Location Summarization

In this section, given a dataset of 504,247 Uber pick ups in Manhattan, New York in April 2014 [41], our goal is to find a set of the most representative locations. This dataset allows us to study the effect of $p$ and $k$ (the size of the largest feasible solution) on the performance of our algorithm.

To do so, the entire area of the given pick ups is covered by $m = 166$ overlapping circular regions of radius $r$ (the centers of these regions provided a 1km-cover of all the area, i.e., for each location in the dataset there was at least one center within a distance of 1km from it), and the algorithm was allowed to choose at most $\ell$ locations out of each one of these regions. One can observe that by using a single matroid for limiting the number of locations chosen within each one of the regions, the above constraint can be expressed as a $p$-matchoid constraint, where $p$ is the maximum number of regions a single location can belong to (notice that $p$ could be much smaller than the total number $m$ of regions).

In order to find a representative set $S$, we use the following monotone submodular objective function: $f(S) = \log \det(I + \alpha K_{S,S})$, where the matrix $K$ encodes the similarities between data points, $K_{S,S}$ is the principal sub-matrix of $K$ indexed by $S$ and $\alpha > 0$ is a regularization parameter [20, 23, 39]. The similarity of two location samples $i$ and $j$ is defined by a Gaussian kernel $K_{i,j} = \exp\left(-d_{i,j}^2/h^2\right)$, where the distance $d_{i,j}$ (in meters) is calculated from the coordinates and $h$ is set to 5000.

In the first experiment, we set the radius of regions to $r = 1.5$km. In this setting, we observed that a point belongs to at most 7 regions; hence, the constraint is a 7-matchoid. For $\ell = 5$, it took 116 seconds[6] (and 693,717 oracle calls) for our algorithm to find a summary of size $k = 153$. Additionally, for $\ell = 10$ and $\ell = 20$ it took 294 seconds (and 1,306,957 oracle calls) and 1004 seconds (and 2,367,389 oracle calls), respectively, for the algorithm to produce summaries of sizes 301 and 541, respectively.

In the second experiment, we set the radius of regions to $r = 2.5$km to investigate the performance of our algorithm on $p$-matchoids with larger values of $p$. In this setting, we observed that a point belongs to at most 17 regions, which made the constraint a 17-matchoid. This time, for $\ell = 5$, it took only 35 seconds (and 296,023 oracle calls) for our algorithm to find a summary of size $k = 54$. Additionally, for $\ell = 10$ and $\ell = 20$ it took 80 seconds (and 526,839 oracle calls) and 176 seconds (and 958,549 oracle calls), respectively, for the algorithm to produce summaries of sizes 106 and 198, respectively. As one can observe, our algorithm scales very well to larger datasets. Also, for $p$-matchoids with larger $p$ (which results in a smaller sampling probability $q$) the performance gets even better.

## 5   Conclusion

We developed a streaming algorithm for submodular maximization by carefully subsampling elements of the data stream. Our algorithm provides the best of three worlds: (i) the tightest approximation guarantees in various settings, including $p$-matchoid and matroid constraints for non-monotone

submodular functions, (ii) minimum memory requirement, and (iii) fewest queries per element. We also experimentally studied the effectiveness of our algorithm.

**Acknowledgements.** The work of Amin Karbasi was supported by AFOSR Young Investigator Award (FA9550-18-1-0160).

## Footnotes

[1]The algorithms of [8] use an offline algorithm for the same problem in a black box fashion, and their approximation ratios depend on the offline algorithm used. The approximation ratios stated here assume the state-of-the-art offline algorithms of [15] which were published only recently, and thus, they are better than the approximation ratios stated by [8].

[2]The memory and query complexities of the algorithm of Mirzasoleiman et al. [33] have been calculated based on the corresponding complexities of the algorithm of [8] for monotone objectives and the properties of the reduction used by [33]. We note that these complexities do not match the memory and query complexities stated by [33] for their algorithm.

[3]`https://github.com/pujols/Video-summarization`

[4]`https://github.com/baharanm/non-mon-stream`

[5]Note that a frame may contain more than one face.

[6]In these experiments, we used a machine powered by Intel i5, 3.2 GHz processor and 16 GB of RAM.

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
