[Supplementary Material · SubmodularStreaming_supplementary.pdf]

## A  Proof of Theorem 4

In this section, we prove Theorem 4. We begin by observing that Algorithm 2 adds an element $u_i$ to its solution if two things happen: (i) $u_i$ is not dismissed due to the random decision and (ii) the marginal contribution of $u_i$ with respect to the current solution is large enough compared to the value of $U_i$. Since checking (ii) requires more resources than checking (i), the algorithm checks (i) first. However, for analyzing the approximation ratio of Algorithm 2, it is useful to assume that (ii) is checked first. Moreover, for the same purpose, it is also useful to assume that the elements that pass (ii) but fail (i) are added to a set $R$. The algorithm obtained after making these changes is given as Algorithm 3. One should note that this algorithm has the same output distribution as Algorithm 2, and thus, the approximation ratio we prove for the first algorithm applies to the second one as well.

---

**Algorithm 3:** Streaming Algorithm for a $p$-Matchoid Constraint (Analysis Version)

---

**1** Let $S_0 \leftarrow \varnothing$ and $R \leftarrow \varnothing$.
**2 for** *every arriving element $u_i$* **do**
**3**     Let $S_i \leftarrow S_{i-1}$.
**4**     Let $U_i \leftarrow$ EXCHANGE-CANDIDATE$(S_{i-1}, u_i)$.
**5**     **if** $f(u_i \mid S_{i-1}) \geq (1 + c) \cdot f(U_i : S_{i-1})$ **then**
**6**        **with** *probability $q$* **do** Let $S_i \leftarrow S_{i-1} \setminus U_i + u_i$.
**7**        **otherwise** Update $R \leftarrow R + u_i$.
**8 return** $S_n$.

---

We now need the following technical observation.

**Observation 6.** *For every two sets $S, T \subseteq \mathcal{N}$, $f(T \mid S \setminus T) \leq f(T : S)$.*

*Proof.* Let us denote the elements of $T$ by $u_{i_1}, u_{i_2}, \ldots, u_{i_{|T|}}$, where $i_1 < i_2 < \cdots < i_{|T|}$. Then,

$$
\begin{aligned}
f(T \mid S \setminus T) &= \sum_{j=1}^{|T|} f(u_{i_j} \mid (S \cup T) \setminus \{u_{i_j}, u_{i_{j+1}} \ldots, u_{i_{|T|}}\}) \\
&\leq \sum_{j=1}^{|T|} f(u_{i_j} \mid S \setminus \{u_{i_j}, u_{i_{j+1}} \ldots, u_n\}) \\
&= \sum_{j=1}^{|T|} f(u_{i_j} \mid S \cap \{u_1, u_2, \ldots, u_{i_j-1}\}) = \sum_{j=1}^{|T|} f(u_{i_j} : S) = f(T : S) \ ,
\end{aligned}
$$

where the inequality follows from the submodularity of $f$. $\qquad\square$

Recall that $A$ is the set of elements that ever appeared in the solution maintained by the algorithm—formally, $A = \bigcup_{i=1}^{n} S_i$. Using the last observation we can prove the following lemma and corollary which show that the elements of $A \setminus S_n$ cannot contribute much to the output solution $S_n$ of Algorithm 3, and thus, their absence from $S_n$ does not make $S_n$ much less valuable than $A$.

**Lemma 7.** $f(A \setminus S_n : S_n) \leq \frac{f(S_n)}{c}$.

*Proof.* Fix an element $u_i \in A$, then

$$
\begin{aligned}
f(S_i) - f(S_{i-1}) &= f(S_{i-1} \setminus U_i + u_i) - f(S_{i-1}) = f(u_i \mid S_{i-1} \setminus U_i) - f(U_i \mid S_{i-1} \setminus U_i) \quad (1) \\
&\geq f(u_i \mid S_{i-1}) - f(U_i : S_{i-1}) \geq c \cdot f(U_i : S_{i-1}) \ ,
\end{aligned}
$$

where the first inequality follows from the submodularity of $f$ and Observation 6, and the second inequality holds since the fact that Algorithm 3 accepted $u_i$ into its solution implies $f(u_i \mid S_{i-1}) \geq (1 + c) \cdot f(U_i : S_{i-1})$.

We now observe that every element of $A \setminus S_n$ must have been removed exactly once from the solution of Algorithm 3, which implies that $\{U_i \mid u_i \in A\}$ is a disjoint partition of $A \setminus S_n$. Using this observation, we get

$$f(A \setminus S_n : S_n) = \sum_{u_i \in A} f(U_i : S_n) \leq \sum_{u_i \in A} \frac{f(S_i) - f(S_{i-1})}{c} = \frac{f(S_n) - f(\varnothing)}{c} \leq \frac{f(S_n)}{c} \ ,$$

where the first inequality follows from Inequality (1), the second equality holds since $S_i = S_{i-1}$ whenever $u_i \notin A$ and the second inequality follows from the non-negativity of $f$. $\qquad\square$

**Corollary 8.** $f(A) \leq \frac{c+1}{c} \cdot f(S_n)$.

*Proof.* Since $S_n \subseteq A$ by definition,

$$f(A) = f(A \setminus S_n \mid S_n) + f(S_n) \leq f(A \setminus S_n : S_n) + f(S_n)$$
$$\leq \frac{f(S_n)}{c} + f(S_n) = \frac{c+1}{c} \cdot f(S_n) \ ,$$

where the first inequality follows from Observation 6 and the second from Lemma 7. $\qquad\square$

Our next goal is to show that the value of the elements of the optimal solution that do not belong to $A$ is not too large compared to the value of $A$ itself. To do so, we need a mapping from the elements of the optimal solution to elements of $A$. Such a mapping is given by Proposition 10. However, before we get to this proposition, let us first present Reduction 9, which simplifies Proposition 10.

**Reduction 9.** *For the sake of analyzing the approximation ratio of Algorithm 3, one may assume that every element $u \in \mathcal{N}$ belongs to* exactly $p$ *out of the $m$ ground sets $\mathcal{N}_1, \mathcal{N}_2, \ldots, \mathcal{N}_m$ of the matroids defining $\mathcal{M}$.*

*Proof.* For every element $u \in \mathcal{N}$ that belongs to the ground sets of only $p' < p$ out of the $m$ matroids $(\mathcal{N}_1, \mathcal{N}_1), (\mathcal{N}_2, \mathcal{N}_2), \ldots, (\mathcal{N}_m, \mathcal{I}_m)$, we can add $u$ to $p - p'$ additional matroids as a free element (i.e., an element whose addition to an independent set always keeps the set independent). On can observe that the addition of $u$ to these matroids does not affect the behavior of Algorithm 3 at all, but makes $u$ obey the technical property of belonging to *exactly* $p$ out of the ground sets $\mathcal{N}_1, \mathcal{N}_2, \ldots, \mathcal{N}_m$. $\quad\square$

From this point on we implicitly make the assumption allowed by Reduction 9. In particular, the proof of Proposition 10 relies on this assumption.

**Proposition 10.** *For every set $T \in \mathcal{I}$ which does not include elements of $R$, there exists a mapping $\phi_T$ from elements of $T$ to multi-subsets of $A$ such that*

- *every element $u \in S_n$ appears at most $p$ times in the multi-sets of $\{\phi_T(u) \mid u \in T\}$.*

- *every element $u \in A \setminus S_n$ appears at most $p - 1$ times in the multi-sets of $\{\phi_T(u) \mid u \in T\}$.*

- *every element $u_i \in T \setminus A$ obeys $f(u_i \mid S_{i-1}) \leq (1 + c) \cdot \sum_{u_j \in \phi_T(u_i)} f(u_j : S_{d(j)-1})$.*

- *every element $u_i \in T \cap A$ obeys $f(u_i \mid S_{i-1}) \leq f(u_j : S_{d(j)-1})$ for every $u_j \in \phi_T(u_i)$, and the multi-set $\phi_T(u_i)$ contains exactly $p$ elements (including repetitions).*

The proof of Proposition 10 is quite long and involves many details, and thus, we defer it to Section A.1. Instead, let us prove now a very useful technical observation. To present this observation we need some additional definitions. Let $Z = \{u_i \in \mathcal{N} \mid f(u_i \mid S_{i-1}) < 0\}$. Additionally, for every $1 \leq i \leq n$, we define

$$d(i) = \begin{cases} 1 + \max\{i \leq j \leq n \mid u_i \in S_j\} & \text{if } u_i \in A \ , \\ i & \text{otherwise} \ . \end{cases}$$

In general, $d(i)$ is the index of the element whose arrival made Algorithm 3 remove $u_i$ from its solution. Two exceptions to this rule are as follows. If $u_i$ was never added to the solution, then $d(i) = i$; and if $u_i$ was never removed from the solution, then $d(i) = n + 1$.

**Observation 11.** *Consider an arbitrary element $u_i \in \mathcal{N}$.*

- If $u_i \notin Z$, then $f(u_i : S_{i'}) \geq 0$ for every $i' \geq i - 1$. In particular, since $d(i) \geq i$, $f(u_i : S_{d(i)-1}) \geq 0$

- $A \cap (R \cup Z) = \varnothing$.

*Proof.* To see why the first part of the observation is true, consider an arbitrary element $u_i \notin Z$. Then,
$$0 \leq f(u_i \mid S_{i-1}) \leq f(u \mid S_{i'} \cap \{u_1, u_2, \ldots, u_{i-1}\}) = f(u : S_{i'}) \ ,$$
where the second inequality follows from the submodularity of $f$ and the inclusion $S_{i'} \cap \{u_1, u_2, \ldots, u_{i-1}\} \subseteq S_{i-1}$ (which holds because elements are only added by Algorithm 3 to its solution at the time of their arrival).

It remains to prove the second part of the observation. Note that Algorithm 3 adds every arriving element to at most one of the sets $A$ and $R$, and thus, these sets are disjoint; hence, to prove the observation it is enough to show that $A$ and $Z$ are also disjoint. Assume towards a contradiction that this is not the case, and let $u_i$ be the first element to arrive which belongs to both $A$ and $Z$. Then,
$$f(u_i \mid S_{i-1}) \geq (1+c) \cdot f(U_i : S_{i-1}) = (1+c) \cdot \sum_{u_j \in U_i} f(u_j : S_{d(j)-1}) \ .$$

To see why that inequality leads to a contradiction, notice its leftmost hand side is negative by our assumption that $u_i \in Z$, while its rightmost hand side is non-negative by the first part of this observation since the choice of $u_i$ implies that no element of $U_i \subseteq S_{i-1} \subseteq A \cap \{u_1, u_2, \ldots, u_{i-1}\}$ can belong to $Z$. $\qquad\square$

Using all the tools we have seen so far, we are now ready to prove Theorem 4. Recall that $OPT$ is an independent set of $\mathcal{M}$ maximizing $f$.

**Theorem 4.** *Assuming* $q^{-1} = (1+c)p + 1$, $\mathbb{E}[f(S_n)] \geq \frac{c}{(1+c)^2 p} \cdot \mathbb{E}[f(A \cup OPT)]$.

*Proof.* Since $S_i \subseteq A$ for every $0 \leq i \leq n$, the submodularity of $f$ guarantees that
$$
\begin{aligned}
f(A \cup OPT) &\leq f(A) + \sum_{u_i \in OPT \setminus (R \cup A)} f(u_i \mid A) + \sum_{u_i \in (OPT \setminus A) \cap R} f(u_i \mid A) \\
&\leq f(A) + \sum_{u_i \in OPT \setminus (R \cup A)} f(u_i \mid S_{i-1}) + \sum_{u_i \in (OPT \setminus A) \cap R} f(u_i \mid S_{i-1}) \\
&\leq \frac{1+c}{c} \cdot f(S_n) + \sum_{u_i \in OPT \setminus (R \cup A)} f(u_i \mid S_{i-1}) + \sum_{u_i \in OPT \cap R} f(u_i \mid S_{i-1}) \ ,
\end{aligned}
$$
where the third inequality follows from Corollary 8 and the fact that $A \cap R = \varnothing$ by Observation 11. Let us now consider the function $\phi_{OPT \setminus R}$ whose existence is guaranteed by Proposition 10 when we choose $T = OPT \setminus R$. Then, the property guaranteed by Proposition 10 for elements of $T \setminus A$ implies
$$\sum_{u_i \in OPT \setminus (R \cup A)} f(u_i \mid S_{i-1}) \leq (1+c) \cdot \sum_{\substack{u_i \in OPT \setminus (R \cup A) \\ u_j \in \phi_{OPT \setminus R}(u_i)}} f(u_j : S_{d(j)-1}) \ .$$

Additionally,
$$
\begin{aligned}
\sum_{\substack{u_i \in OPT \setminus (R \cup A) \\ u_j \in \phi_{OPT \setminus R}(u_i)}} f(u_j : S_{d(j)-1}) + p \cdot \sum_{u_i \in OPT \cap A} f(u_i \mid S_{i-1}) &\leq \sum_{\substack{u_i \in OPT \setminus R \\ u_j \in \phi_{OPT \setminus R}(u_i)}} f(u_j : S_{d(j)-1}) \\
&\leq p \cdot \sum_{u_j \in S_n} f(u_j : S_n) + (p-1) \cdot \sum_{u_j \in A \setminus S_n} f(u_j : S_{d(j)-1}) \\
&\leq p \cdot f(S_n) + \frac{p-1}{c} \cdot f(S_n) = \frac{(1+c) \cdot p - 1}{c} \cdot f(S_n) \ ,
\end{aligned}
$$
where the first inequality follows from the properties guaranteed by Proposition 10 for elements of $T \cap A$ (note that the sets $OPT \setminus (R \cup A)$ and $OPT \cap A$ are a disjoint partition of $OPT \setminus R$ by

Observation 11) and the second inequality follows from the properties guaranteed by Proposition 10 for elements of $A \setminus S_n$ and $S_n$ because every element $u_i$ in the multisets produced by $\phi_{OPT \setminus R}$ belongs to $A$, and thus, obeys $f(u_i : S_{d(i)-1}) \geq 0$ by Observation 11. Finally, the last inequality follows from Lemma 7 and the fact that $f(u_j : S_{d(j)-1}) \leq f(u_j : S_n)$ for every $1 \leq j \leq n$. Combining all the above inequalities, we get

$$f(A \cup OPT) \leq \frac{1+c}{c} \cdot f(S_n) +$$

$$(1+c) \cdot \left[ \frac{(1+c) \cdot p - 1}{c} \cdot f(S_n) - p \cdot \sum_{u_i \in OPT \cap A} f(u_i \mid S_{i-1}) \right] + \sum_{u_i \in OPT \cap R} f(u_i \mid S_{i-1})$$

$$= \frac{(1+c)^2 \cdot p}{c} \cdot f(S_n) - (1+c)p \cdot \sum_{u_i \in OPT \cap A} f(u_i \mid S_{i-1}) + \sum_{u_i \in OPT \cap R} f(u_i \mid S_{i-1}) \ . \qquad (2)$$

By the linearity of expectation, to prove the theorem it only remains to show that the expectations of the last two terms on the rightmost hand side of Inequality (2) are equal. This is our objective in the rest of this proof. Consider an arbitrary element $u_i \in OPT$. When $u_i$ arrives, one of two things happens. The first option is that Algorithm 3 discards $u_i$ without adding it to either its solution or to $R$. The other option is that Algorithm 3 adds $u_i$ to its solution (and thus, to $A$) with probability $q$, and to $R$ with probability $1 - q$. The crucial observation here is that at the time of $u_i$'s arrival the set $S_{i-1}$ is already determined, and thus, this set is independent of the decision of the algorithm to add $u$ to $A$ or to $R$; which implies the following equality (given an event $\mathcal{E}$, we use here $\mathbf{1}[\mathcal{E}]$ to denote an indicator for it).

$$\frac{\mathbb{E}[\mathbf{1}[u_i \in A] \cdot f(u_i \mid S_{i-1})]}{q} = \frac{\mathbb{E}[\mathbf{1}[u_i \in R] \cdot f(u_i \mid S_{i-1})]}{1-q} \ .$$

Rearranging the last equality, and summing it up over all elements $u_i \in OPT$, we get

$$\frac{1-q}{q} \cdot \mathbb{E}\left[ \sum_{u_i \in OPT \cap A_n} f(u_i \mid S_{i-1}) \right] = \mathbb{E}\left[ \sum_{u_i \in OPT \cap R} f(u_i \mid S_{i-1}) \right] \ .$$

Recall that we assume $q^{-1} = (c+1)p + 1$, which implies $(1-q)/q = q^{-1} - 1 = (c+1)p$. Plugging this equality into the previous one completes the proof that the expectations of the last two terms on the rightmost hand side of Inequality (2) are equal. $\qquad \square$

## A.1 Proof of Proposition 10

In this section we prove Propsition 10. Let us first restate the proposition itself.

**Proposition 10.** *For every set $T \in \mathcal{I}$ which does not include elements of $R$, there exists a mapping $\phi_T$ from elements of $T$ to multi-subsets of $A$ such that*

- *every element $u \in S_n$ appears at most $p$ times in the multi-sets of $\{\phi_T(u) \mid u \in T\}$.*

- *every element $u \in A \setminus S_n$ appears at most $p - 1$ times in the multi-sets of $\{\phi_T(u) \mid u \in T\}$.*

- *every element $u_i \in T \setminus A$ obeys $f(u_i \mid S_{i-1}) \leq (1+c) \cdot \sum_{u_j \in \phi_T(u_i)} f(u_j : S_{d(j)-1})$.*

- *every element $u_i \in T \cap A$ obeys $f(u_i \mid S_{i-1}) \leq f(u_j : S_{d(j)-1})$ for every $u_j \in \phi_T(u_i)$, and the multi-set $\phi_T(u_i)$ contains exactly $p$ elements (including repetitions).*

We begin the proof of Proposition 10 by constructing $m$ graphs, one for every one of the matroids defining $\mathcal{M}$. For every $1 \leq \ell \leq m$, the graph $G_\ell$ contains two types of vertices: its internal vertices are the elements of $A \cap \mathcal{N}_\ell$, and its external vertices are the elements of $\{u_i \in \mathcal{N}_\ell \setminus (R \cup A) \mid (S_{i-1} + u_i) \cap \mathcal{N}_\ell \notin \mathcal{I}_\ell\}$. Informally, the external elements of $G_\ell$ are the elements of $\mathcal{N}_\ell$ which were rejected upon arrival by Algorithm 3 and the matroid $\mathcal{M}_\ell = (\mathcal{N}_\ell, \mathcal{I}_\ell)$ can be (partially) blamed for this rejection. The arcs of $G_\ell$ are created using the following iterative process that creates some arcs of $G_\ell$ in response to every arriving element. For every $1 \leq i \leq n$, consider the element $x_\ell$ selected by the execution of EXCHANGE-CANDIDATE on the element $u_i$ and the set $S_{i-1}$. From

this point on we denote this element by $x_{i,\ell}$. If no $x_{i,\ell}$ element was selected by the above execution of EXCHANGE-CANDIDATE, or $u_i \in R$, then no $G_\ell$ arcs are created in response to $u_i$. Otherwise, let $C_{i,\ell}$ be the single cycle of the matroid $\mathcal{M}_\ell$ in the set $(S_{i-1} + u_i) \cap \mathcal{N}_\ell$—there is exactly one cycle of $\mathcal{M}_\ell$ in this set because $S_{i-1}$ is independent, but $(S_{i-1} + u_i) \cap \mathcal{N}_\ell$ is not independent in $\mathcal{M}_\ell$. One can observe that $C_{i,\ell} - u_i$ is equal to the set $X_\ell$ in the above-mentioned execution of EXCHANGE-CANDIDATE, and thus, $x_{i,\ell} \in C_{i,\ell}$. We now denote by $u'_{i,\ell}$ the vertex out of $\{u_i, x_{i,\ell}\}$ that does not belong to $S_i$—notice that there is exactly one such vertex since $x_{i,\ell} \in U_i$, which implies that it appears in $S_i$ if $S_i = S_{i-1}$ and does not appear in $S_i$ if $S_i = S_{i-1} \setminus U_i + u_i$. Regardless of the node chosen as $u'_i$, the arcs of $G_\ell$ created in response to $u_i$ are all the possible arcs from $u'_{i,\ell}$ to the other vertices of $C_{i,\ell}$. Observe that these are valid arcs for $G_\ell$ in the sense that their endpoints (i.e., the elements of $C_{i,\ell}$) are all vertices of $G_\ell$—for the elements of $C_{i,\ell} - u_i$ this is true since $C_{i,\ell} - u_i \subseteq S_{i-1} \cap \mathcal{N}_\ell \subseteq A \cap \mathcal{N}_\ell$, and for the element $u_i$ this is true since the existence of $x_{i,\ell}$ implies $(S_{i-1} + u_i) \cap \mathcal{N}_\ell \notin \mathcal{I}_\ell$.

Some properties of $G_\ell$ are given by the following observation. Given a graph $G$ and a vertex $u$, we denote by $\delta_G^+(u)$ the set of vertices to which there is a direct arc from $u$ in $G$.

**Observation 12.** *For every $1 \leq \ell \leq m$,*

- *every non-sink vertex $u$ of $G_\ell$ is spanned by the set $\delta_{G_\ell}^+(u)$.*

- *for every two indexes $1 \leq i, j \leq n$, if $u'_{i,\ell}$ and $u'_{j,\ell}$ both exist and $i \neq j$, then $u'_{i,\ell} \neq u'_{j,\ell}$.*

- *$G_\ell$ is a directed acyclic graph.*

*Proof.* Consider an arbitrary non-sink node $u$ of $G_\ell$. Since there are arcs leaving $u$, $u$ must be equal to $u'_{i,\ell}$ for some $1 \leq i \leq n$. This implies that $u$ belongs to the cycle $C_{i,\ell}$, and that there are arcs from $u$ to every other vertex of $C_{i,\ell}$. Thus, $u$ is spanned by the vertices of $\delta_{G_\ell}^+(u) \supseteq C_{i,\ell} - u$ because the fact that $C_{i,\ell}$ is a cycle containing $u$ implies that $C_{i,\ell} - u$ spans $u$. This completes the proof of the first part of the observation.

Let us prove now a very useful technical claim. Consider an index $1 \leq i \leq n$ such that $u'_i$ exists, and let $j$ be an arbitrary value $i < j \leq n$. We will prove that $u'_i$ does not belong to $C_{j,\ell}$. By definition, $u'_i$ is either $u_i$ or the vertex $x_{i,\ell}$ that belongs to $S_{i-1}$, and thus, arrived before $u_i$ and is not equal to $u_j$; hence, in neither case $u'_i \neq u_j$. Moreover, combining the fact that $u'_i$ is either $u_i$ or arrived before $u_i$ and the observation that $u'_i$ is never a part of $S_i$, we get that $u'_i$ cannot belong to $S_j \supseteq C_{j,\ell} - u_j$, which implies the claim together without previous observation that $u'_i \neq u_j$.

The technical claim that we proved above implies the second part of the lemma, namely that for every two indexes $1 \leq i, j \leq n$, if $u'_{i,\ell}$ and $u'_{j,\ell}$ both exist and $i \neq j$, then $u'_{i,\ell} \neq u'_{j,\ell}$. To see why that is the case, assume without loss of generality $i < j$. Then, the above technical claim implies that $u'_{i,\ell} \notin C_{j,\ell}$, which implies $u'_{i,\ell} \neq u'_{j,\ell}$ because $u'_{j,\ell} \in C_{j,\ell}$.

At this point, let us assume towards a contradiction that the third part of the observation is not true, i.e., that there exists a cycle $L$ in $G_\ell$. Since every vertex of $L$ has a non-zero out degree, every such vertex must be equal to $u'_{i,\ell}$ for some $1 \leq i \leq n$. Thus, there must be indexes $1 \leq i_1 < i_2 \leq n$ such that $L$ contains an arc from $u'_{i_2,\ell}$ to $u'_{i_1,\ell}$. Since we already proved that $u'_{i_2,\ell}$ cannot be equal to $u'_{j,\ell}$ for any $j \neq i_2$, the arc from $u'_{i_2,\ell}$ to $u'_{i_1,\ell}$ must have been created in response to $u_{i_2}$, hence, $u'_{i_1,\ell} \in C_{i_2,\ell}$, which contradicts the technical claim we have proved. $\square$

One consequence of the properties of $G_\ell$ proved by the last observation is given by the following lemma. A slightly weaker version of this lemma was proved implicitly by [42], and was stated as an explicit lemma by [8].

**Lemma 13.** *Consider an arbitrary directed acyclic graph $G = (V, E)$ whose vertices are elements of some matroid $\mathcal{M}'$. If every non-sink vertex $u$ of $G$ is spanned by $\delta_G^+(u)$ in $\mathcal{M}'$, then for every set $S$ of vertices of $G$ which is independent in $\mathcal{M}'$ there must exist an injective function $\psi_S$ such that, for every vertex $u \in S$, $\psi_S(u)$ is a sink of $G$ which is reachable from $u$.*

*Proof.* Let us define the *width* of a set $S$ of vertices of $G$ as the number of arcs that appear on some path starting at a vertex of $S$ (more formally, the width of $S$ is the size of the set $\{e \in E \mid$

there is a path in $G$ that starts in a vertex of $S$ and includes $e$}). We prove the lemma by induction of the width of $S$. First, consider the case that $S$ is of width 0. In this case, the vertices of $S$ cannot have any outgoing arcs because such arcs would have contributed to the width of $S$, and thus, they are all sinks of $G$. Thus, the lemma holds for the trivial function $\psi_S$ mapping every element of $S$ to itself. Assume now that the width $w$ of $S$ is larger than 0, and assume that the lemma holds for every set of width smaller than $w$. Let $u$ be a non-sink vertex of $S$ such that there is no path in $G$ from any other vertex of $S$ to $u$. Notice that such a vertex must exist since $G$ is acyclic. By the assumption of the lemma, $\delta^+(u)$ spans $u$. In contrast, since $S$ is independent, $S - u$ does not span $u$, and thus, there must exist an element $v \in \delta^+(u) \setminus S$ such that the set $S' = S - u + v$ is independent.

Let us explain why the width of $S'$ must be strictly smaller than the width of $S$. First, consider an arbitrary arc $e$ which is on a path starting at a vertex $u' \in S'$. If $u' \in S$, then $e$ is also on a path starting in a vertex of $S$. On the other hand, if $u' \notin S$, then $u'$ must be the vertex $v$. Thus, $e$ must be on a path $P$ starting in $v$. Adding $uv$ to the beginning of the path $P$, we get a path from $u$ which includes $e$. Hence, in conclusion, we have got that every arc $e$ which appears on a path starting in a vertex of $S'$ (and thus, contributes to the width of $S'$) also appears on a path starting in a vertex of $S$ (and thus, also contributes to the width of $S$); which implies that the width of $S'$ is not larger than the width of $S$. To see that the width of $S'$ is actually strictly smaller than the width of $S$, it only remains to find an arc which contributes to the width of $S$, but not to the width of $S'$. Towards this goal, consider the arc $uv$. Since $u$ is a vertex of $S$, the arc $uv$ must be on some path starting in $u$ (for example, the path including only this arc), and thus, contributes to the width of $S$. Assume now towards a contradiction that $uv$ contributes also to the width of $S'$, i.e., that there is a path $P$ starting at a vertex $w \in S'$ which includes $uv$. If $w = v$, then this leads to a contradiction since it implies the existence of a cycle in $G$. On the other hand, if $w \neq v$, then this implies a path in $G$ from a vertex $w \neq u$ of $S$ to $u$, which contradicts the definition of $u$. This completes the proof that the width of $S'$ is strictly smaller than the width of $S$.

Using the induction hypothesis, we now get that there exists an injective function $\psi_{S'}$ mapping every vertex of $S'$ to a sink of $G$. Using $\psi_{S'}$, we can define $\psi_S$ as follows. For every $w \in S$,

$$\psi_S(w) = \begin{cases} \psi_{S'}(v) & \text{if } w = u \ , \\ \psi_{S'}(w) & \text{otherwise} \ . \end{cases}$$

Since $u$ appears in $S$ but not in $S'$, and $v$ appears in $S'$ but not in $S$, the injectiveness of $\psi_S$ follows from the injectiveness of $\psi_{S'}$. Moreover, $\psi_S$ clearly maps every vertex of $S$ to a sink of $G$ since $\psi_{S'}$ maps every vertex of $S'$ to such a sink. Finally, one can observe that $\psi_S(w)$ is reachable from $w$ for every $w \in S$ because $\psi_S(u) = \psi_{S'}(v)$ is reachable from $v$ by the definition of $\psi_{S'}$, and thus, also from $u$ due to the existence of the arc $uv$. $\qquad\square$

For every $1 \leq \ell \leq m$, let $T_\ell$ be the set of elements of $T$ that appear as vertices of $G_\ell$. Since $T$ is independent and $T_\ell$ contains only elements of $\mathcal{N}_\ell$, Observation 12 and Lemma 13 imply together the existence of an injective function $\psi_{T_\ell}$ mapping the elements of $T_\ell$ to sink vertices of $G_\ell$. We can now define the function $\phi_T$ promised by Proposition 10. For every element $u \in T$, the function $\phi_T$ maps $u$ to the multi-set $\{\psi_{T_\ell}(u) \mid 1 \leq \ell \leq m \text{ and } u \in T_\ell\}$, where we assume that repetitions are kept when the expression $\psi_{T_\ell}(u)$ evaluates to the same element for different choices of $\ell$. Let us explain why the elements in the multi-sets produced by $\phi_T$ are indeed all elements of $A$, as is required by the proposition. Consider an element $u_i \notin A$, and let us show that it does not appear in the range of $\psi_{T_\ell}$ for any $1 \leq \ell \leq m$. If $u_i$ does not appear as a vertex in $G_\ell$, then this is obvious. Otherwise, the fact that $u_i \notin A$ implies $u'_{i,\ell} = u_i$, and thus, the arcs of $G_\ell$ created in response to $u_i$ are arcs leaving $u_i$, which implies that $u_i$ is not a sink of $G_\ell$, and hence, does not appear in the range of $\psi_{T_\ell}$.

Recall that every element $u \in \mathcal{N}$ belongs to at most $p$ out of the ground sets $\mathcal{N}_1, \mathcal{N}_2, \ldots, \mathcal{N}_m$, and thus, is a vertex in at most $p$ out of the graphs $G_1, G_2, \ldots, G_m$. Since $\psi_{T_\ell}$ maps every element to vertexes of $G_\ell$, this implies that $u$ is in the range of at most $p$ out of the functions $\psi_{T_1}, \psi_{T_2}, \ldots, \psi_{T_m}$. Moreover, since these functions are injective, every one of these functions that have $u$ in its range maps at most one element to $u$. Thus, the multi-sets produced by $\phi_T$ contain $u$ at most $p$ times. Since this is true for every element of $\mathcal{N}$, it is true in particular for the elements of $S_n$, which is the first property of $\phi_T$ that we needed to prove.

Consider now an element $u \in A \setminus S_n$. Our next objective is to prove that $u$ appears at most $p - 1$ times in the multi-sets produced by $\phi_T$, which is the second property of $\phi_T$ that we need to prove.

Above, we proved that $u$ appears at most $p$ times in these multi-sets by arguing that every such appearance must be due to a function $\psi_{T_\ell}$ that has $u$ in its range, and that the function $\psi_{T_\ell}$ can have this property only for the $p$ values of $\ell$ for which $u \in \mathcal{N}_\ell$. Thus, to prove that $u$ in fact appears only $p - 1$ times in the multi-sets produced by $\phi_T$, it is enough to argue that there exists a value $\ell$ such that $e \in \mathcal{N}_\ell$, but $\psi_{T_\ell}$ does not have $u$ in its range. Let us prove that this follows from the membership of $u$ in $A \setminus S_n$. Since $u$ was removed from the solution of Algorithm 3 at some point, there must be some index $1 \leq i \leq n$ such that both $u \in U_i$ and $u_i$ was added to the solution of Algorithm 3. Since $u \in U_i$, there must be a value $1 \leq \ell \leq m$ such that $u = x_{i,\ell}$, and since $u_i$ was added to the solution of Algorithm 3, $u'_{i,\ell} = x_{i,\ell}$. These equalities imply together that there are arcs leaving $u$ in $G_\ell$ (which were created in response to $u_i$). Thus, the function $\psi_{T_\ell}$ does not map any element to $u$ because $u$ is not a sink of $G_\ell$, despite the fact that $u \in \mathcal{N}_\ell$.

To prove the other guaranteed properties of $\phi_T$, we need the following lemma.

**Lemma 14.** *Consider two vertices $u_i$ and $u_j$ such that $u_j$ is reachable from $u_i$ in $G_\ell$. If $u_i \in A$, then $f(u_i : S_{d(i)-1}) \leq f(u_j : S_{d(j)-1})$, otherwise, $f(x_{i,\ell} : S_{i-1}) \leq f(u_j : S_{d(j)-1})$.*

*Proof.* We begin by proving the special case of the lemma in which $u_i \in A$ (i.e., is an internal vertex of $G_\ell$) and there is a direct arc from $u_i$ to $u_j$. The existence of this arc implies that there is some value $1 \leq h \leq n$ such that $u'_{h,\ell} = u_i$ and $u_j \in C_{h,\ell}$. Since $u_i$ is internal, it cannot be equal be to $u_h$ because this would have implied that $u_h$ was rejected immediately by Algorithm 3, and is thus, not internal. Thus, $u_i = x_{h,\ell}$. Recall now that $C_{h,\ell} - u_h$ is equal to the set $X_\ell$ chosen by EXCHANGE-CANDIDATE when it is executed with the element $u_h$ and the set $S_{h-1}$. Thus, the fact that $u_i = x_{h,\ell}$ and the way $x_{h,\ell}$ is chosen out of $X_\ell$ implies that whenever $u_j \neq u_h$ we have

$$f(u_i : S_{d(i)-1}) = f(u_i : S_{h-1}) \leq f(u_j : S_{h-1}) \leq f(u_j : S_{d(j)-1}) \ ,$$

where the equality holds since $u'_{h,\ell} = u_i$ implies $d(i) = h$ and the last inequality holds since $f(u_j : S_{r-1})$ is a non-decreasing function of $r$ when $r \geq j$ and the membership of $u_j$ in $C_{h,\ell}$ implies $j \leq h \leq d(j)$.

It remains to consider the case $u_j = u_h$. In this case, the fact that $u_j = u_h$ is accepted into the solution of Algorithm 3 implies

$$\begin{aligned} f(u_j : S_{d(j)-1}) &\geq f(u_j : S_{j-1}) = f(u_j \mid S_{j-1} \cap \{u_1, u_2, \ldots, u_{j-1}\}) = f(u_j \mid S_{j-1}) \\ &= f(u_h \mid S_{h-1}) \geq (1+c) \cdot f(U_h : S_{h-1}) \geq f(U_h : S_{h-1}) \\ &\geq f(x_{h,\ell} : S_{h-1}) = f(u_i : S_{h-1}) = f(u_i : S_{d(i)-1}) \ , \end{aligned}$$

where the first inequality holds since $d(j) \geq j$ by definition, the last equality holds since $u'_{h,\ell} = u_i$ implies $d(i) = h$ and the two last inequalities follow from the fact that the elements of $U_h \subseteq A$ do not belong to $Z$ by Observation 11, which implies (again, by Observation 11) that $f(u : S_{h-1}) \geq 0$ for every $u \in U_h$. This completes the proof of the lemma for the special case that $u_i \in A$ and there is a direct arc from $u_i$ to $u_j$.

Next, we prove that no arc of $G_\ell$ goes from an internal vertex to an external one. Assume this is not the case, and that there exists an arc $uv$ of $G_\ell$ from an internal vertex $u$ to an external vertex $v$. By definition, there must be a value $1 \leq h \leq n$ such that $v$ belongs to the cycle $C_{h,\ell}$ and $u'_{h,\ell} = u$. The fact that $u$ is an internal vertex implies that $u_h$ must have been accepted by Algorithm 3 upon arrival because otherwise we would have gotten $u = u'_{h,\ell} = u_h$, which implies that $u$ is external, and thus, leads to a contradiction. Consequently, we get $C_{h,\ell} \subseteq A$ because every element of $C_{h,\ell}$ must either be $u_h$ or belong to $S_{h-1}$. In particular, $v \in A$, which contradicts our assumption that $v$ is an external vertex.

We are now ready to prove the lemma for the case $u_i \in A$ (even when there is no direct arc in $G_\ell$ from $u_i$ to $u_j$). Consider some path $P$ from $u_i$ to $u_j$, and let us denote the vertices of this path by $u_{r_0}, u_{r_1}, \ldots, u_{r_{|P|}}$. Since $u_i$ is an internal vertex of $G_\ell$ and we already proved that no arc of $G_\ell$ goes from an internal vertex to an external one, all the vertices of $P$ must be internal. Thus, by applying the special case of the lemma that we have already proved to every pair of adjacent vertices along the path $P$, we get that the expression $f(u_{r_k} : S_{d(r_k)-1})$ is a non-decreasing function of $k$, and in particular,

$$f(u_i : S_{d(i)-1}) = f(u_{r_0} : S_{d(r_0)-1}) \leq f(u_{r_k} : S_{d(r_k)-1}) = f(u_j : S_{d(j)-1}) \ .$$

It remains to prove the lemma for the case $u_i \notin A$. Let $u_h$ denote the first vertex on some path from $u_i$ to $u_j$ in $G_\ell$. Since $u_i \notin A$, we get that $u'_{i,\ell} = u_i$, which implies that the arcs of $G_\ell$ that were created in response to $u_i$ go from $u_i$ to the vertices of $C_{i,\ell} - u_i$. Since Observation 12 guarantees that $u_i = u'_{i,\ell} \neq u'_{j,\ell}$ for every value $1 \leq j \leq n$ which is different from $i$, there cannot be any other arcs in $G_\ell$ leaving $u_i$, and thus, the existence of an arc from $u_i$ to $u_h$ implies $u_h \in C_{i,\ell} - u_i$. Recall now that $C_{i,\ell} - u_i$ is equal to the set $X_\ell$ in the execution of EXCHANGE-CANDIDATE corresponding to the element $u_i$ and the set $S_{i-1}$, and thus, by the definition of $x_{i,\ell}$, $f(x_{i,\ell} : S_{i-1}) \leq f(u_h : S_{i-1})$. Additionally, as an element of $C_{i,\ell} - u_i$, $u_h$ must be a member of $S_{i-1} \subseteq A$, and thus, by the part of the lemma we have already proved, we get $f(u_h : S_{d(h)-1}) \leq f(u_j : S_{d(j)-1})$ because $u_j$ is reachable from $u_h$. Combining the two inequalities we have proved, we get

$$f(x_{i,\ell} : S_{i-1}) \leq f(u_h : S_{i-1}) \leq f(u_h : S_{d(h)-1}) \leq f(u_j : S_{d(j)-1}) \ ,$$

where the second inequality holds since the fact that $u_h \in C_{i,\ell} - u_i \subseteq S_{i-1}$ implies $d(h) \geq i$.  $\square$

Consider now an arbitrary element $u_i \in T \setminus A$. Let us denote by $u_{r_\ell}$ the element $u_{r_\ell} = \psi_{T_\ell}(u_i)$ if it exists, and recall that this element is reachable from $u_i$ in $G_\ell$. Thus, the fact that $u_i$ is not in $A$ implies

$$f(u_i \mid S_{i-1}) \leq (1+c) \cdot \sum_{u \in U_i} f(u : S_{i-1}) = (1+c) \cdot \sum_{\substack{1 \leq \ell \leq m \\ (S_{i-1}+u_i) \cap \mathcal{N}_\ell \notin \mathcal{I}_\ell}} f(x_{i,\ell} : S_{i-1})$$

$$\leq (1+c) \cdot \sum_{\substack{1 \leq \ell \leq m \\ (S_{i-1}+u_i) \cap \mathcal{N}_\ell \notin \mathcal{I}_\ell}} f(u_{r_\ell} : S_{d(r_\ell)}) = (1+c) \cdot \sum_{u_j \in \phi_T(u_i)} f(u_j : S_{d(j)}) \ ,$$

where the inequality follows from Lemma 14 and the last equality holds since the values of $\ell$ for which $(S_{i-1} + u_i) \cap \mathcal{N}_\ell \notin \mathcal{I}_\ell$ are exactly the values for which $u_i \in T_\ell$, and thus, they are all also exactly the values for which the multi-set $\phi_T(u_i)$ includes the value of $\psi_{T_\ell}(u_i)$. This completes the proof of the third property of $\phi_T$ that we need to prove.

Finally, consider an arbitrary element $u_i \in A \cap T$. Every element $u_j \in \phi_T(u_i)$ can be reached from $u_i$ in some graph $G_\ell$, and thus, by Lemma 14,

$$f(u_i \mid S_{i-1}) = f(u_i \mid S_{i-1} \cap \{u_1, u_2, \ldots, u_{i-1}\}) = f(u_i : S_{i-1})$$
$$\leq f(u_i : S_{d(i)-1}) \leq f(u_j : S_{d(j)-1}) \ ,$$

where the first inequality holds since $d(i) \geq i$ by definition and $f(u_i : S_{r-1})$ is a non-decreasing function of $r$ for $r \geq i$. Additionally, we observe that $u_i$, as an element of $T \cap A$, belongs to $T_\ell$ for every value $1 \leq \ell \leq m$ for which $u_i \in \mathcal{N}_\ell$, and thus, the size of the multi-set $\phi_T(u_i)$ is equal to the number of ground sets out of $\mathcal{N}_1, \mathcal{N}_2, \ldots, \mathcal{N}_m$ that include $u_i$. Since we assume by Reduction 9 that every element belongs to exactly $p$ out of these ground sets, we get that the multi-set $\phi_T(u_i)$ contains exactly $p$ elements (including repetitions), which completes the proof of Proposition 10.