[Reviews · NeurIPS 2018]

Reviewer 1



The paper studies the problem of maximizing a submodular function on a stream subject to a p-matchoid constraint. A p-matchoid constraint is a system of m matroid constraints such that each element participates in at most p of those matroid constraints. Let k be the maximum size of the feasible sets. The main result of the paper is an algorithm for non-monotone submodular functions that gets approximation 4p+2, improving the previous result of 4p+4*sqrt(p)+1. A by-product of their result is an algorithm for monotone functions that gets the same approximation as previous works (4p) but with faster running time. The improvement comes from the idea that for every element, one can discard it with probability 2p/(2p+1). Even though we lose because we might accidentally remove elements in OPT, this can be balanced with the gain we get when including those elements (some of these terms were ignored before). As the matchoid constraint is quite general, the charging argument when the algorithm exchanges existing elements in the solution for a new element is fairly complicated and I have not checked the proof carefully. Overall the paper has a nice idea to use the gain from having elements in OPT added to the solution at some point to counteract removing elements randomly and achieving the same approximation as before. One downside is that I am not sure why one has to use streaming algorithm for the mentioned applications. It seems in all cases, the data is already stored and not coming online. They also seem small enough to fit in memory at once.

Reviewer 2



This is a clearly written and well argued paper that improves the known state of the art algorithms for the specific problem of streaming submodular optimization subject to multiple matroid (or matchoid) constraints. The authors improve the running time and either match or improve the approximation factors of known algorithms. The experimental results can be significantly strengthened. One downside of the work is that the theoretical improvement is relatively small, however in my mind this is outweighed by the fact that the algorithm is clean, seemingly universal (subsuming previous work), and practical. On the technical side, the main contribution is Theorem 4, which gives a parametrized approximation guarantee for the Sample-Streaming algorithm. This is then combined with known results, and tuned settings for the parameters to obtain the stated results for monotone and non-monotone functions. The proof is involved, it would have been good to provide a high level intuition picture in the Appendix (or if possible in the main body) of why this approach succeeds where previous methods have failed. More notes: - Spell check -- l176 (independence), l248 (diverse), - In the experimental section (line 265), it is worth noting explicitly that the proposed algorithm performs _worse_ than the SeqDPP baseline on OVP videos, and on half of the YouTube videos. A sentence is also warranted to describe the performance versus local search, which has a strictly worse approximation ratio, but performs comparably here. - While it's nice to see absolute running time comparisons in Figure 1c, a much more meaningful metric would be the number of oracle calls, since this is precisely what the proposed method is trying to optimized. Moreover, such a metric would be independent of machine speed, or programming prowess, and can be used as a basis for comparison in the future. I strongly urge the authors to provide such a plot. Rebuttal Question: how does the number of evaluations scale between different methods? - In my mind Section 4.2 should either be cut, or augmented with a stronger baseline. As mentioned above, simply reporting running time numbers without a baseline is not very meaningful, and future researchers would not be able to compare their methods against them.

Reviewer 3



The paper proposes a subsampling streaming submodular maximization algorithm. The main improvement is reducing the number of queries per element. Overall, I recommend the paper to be accepted. I think the analysis of the paper is interesting. I first wondered how the probability q affects the approximation factors. But Eq (2) shows that it only affects the non-dominating terms. Therefore, we can choose it as large as possible. I checked all the proofs except Proposition 10 (I only skimmed Prop. 10), and found no flaws. [minor] Theorem 1. - the formula needs a close parenthesis. line 429 - the second line, n will be u_{i_{|T|}} - the first equality in the third line will be the inequality.